# An Innovative Thermal Imaging Prototype for Precise Breast Cancer Detection: Integrating Compression Techniques and Classification Methods

**DOI:** 10.3390/bioengineering11080764

**Published:** 2024-07-29

**Authors:** Khaled S. Ahmed, Fayroz F. Sherif, Mohamed S. Abdallah, Young-Im Cho, Shereen M. ElMetwally

**Affiliations:** 1Bio-Medical Department, Benha University, Benha 13518, Egypt; khaled.sayed@bhit.bu.edu.eg; 2Computers and Systems Department, Electronics Research Institute (ERI), Cairo 11843, Egypt; fayroz_farouk@eri.sci.eg; 3Informatics Department, Electronics Research Institute (ERI), Cairo 11843, Egypt; 4AI Laboratory, DeltaX Co., Ltd., Seoul 08213, Republic of Korea; 5Department of Computer Engineering, Gachon University, Seongnam 13415, Republic of Korea; 6Systems and Biomedical Engineering Department, Cairo University, Giza 12613, Egypt; sh.elmetwally@eng1.cu.edu.eg

**Keywords:** thermal imaging, thermography device, breast cancer detection, classification

## Abstract

Breast cancer detection at an early stage is crucial for improving patient survival rates. This work introduces an innovative thermal imaging prototype that incorporates compression techniques inspired by mammography equipment. The prototype offers a radiation-free and precise cancer diagnosis. By integrating compression and illumination methods, thermal picture quality has increased, and the accuracy of classification has improved. Essential components of the suggested thermography device include an equipment body, plates, motors, pressure sensors, light sources, and a thermal camera. We created a 3D model of the gadget using the SolidWorks software 2020 package. Furthermore, the classification research employed both cancer and normal images from the experimental results to validate the efficacy of the suggested system. We employed preprocessing and segmentation methods on the obtained dataset. We successfully categorized the thermal pictures using various classifiers and examined their performance. The logistic regression model showed excellent performance, achieving an accuracy of 0.976, F1 score of 0.977, precision of 1.000, and recall of 0.995. This indicates a high level of accuracy in correctly classifying thermal abnormalities associated with breast cancer. The proposed prototype serves as a highly effective tool for conducting initial investigations into breast cancer detection, offering potential advancements in early-stage diagnosis, and improving patient survival rates.

## 1. Introduction

Cancer refers to a group of diseases characterized by the development of malignant cells within the human body because of genetic alterations. These cells undergo uncontrolled division throughout development, spreading throughout the organs and often leading to fatal consequences. Cardiovascular diseases are the only cause of death that surpasses cancer in terms of worldwide mortality [1]. Breast cancer, which has a mortality rate of 20.1 per 100,000 women annually, is the most common cancer among women. Statistically, 95% of breast cancer cases are classified as adenocarcinomas, and 55% of all patients may progress to invasive stages. However, effective treatment is possible in around 70–80% of cases if the cancer is discovered early [2].

The origins of breast cancer may be traced to over 3500 years ago when it was initially documented by the ancient Egyptians in 1500 B.C. [3]. Currently, breast cancer ranks as the second most common form of cancer and is a primary contributor to most cancer-related fatalities among women in the United States [2]. The American Cancer Society estimates that around 310,720 women will receive a diagnosis of breast cancer in 2024, and 42,250 women are expected to die from the disease in the United States [4]. Prompt detection of the illness is essential for successful therapy and a favorable outlook, as patients with smaller tumors at the time of diagnosis had a much-reduced likelihood of mortality and a greater rate of survival [2].

Early detection is essential in the context of breast cancer. By identifying the disease at an earlier, more treatable stage, healthcare providers can significantly improve the chances of survival and quality of life for their patients. Numerous studies have consistently shown a strong correlation between the size of the tumor at the time of diagnosis and the patient’s long-term prognosis. A study in [5] revealed that patients diagnosed with smaller tumors (less than 2 cm) were significantly less likely to die than those with larger tumors [5]. Specifically, the 5-year survival rate for patients with tumors less than 1 cm was 90%, while the 5-year survival rate for those with tumors between 1 and 2 cm was 78%. In contrast, the 5-year survival rate dropped significantly to just 59% for patients diagnosed with tumors larger than 2 cm [5].

The 5-year relative survival rate for localized breast cancer (stage I) was 99%, but only 86% for regional-stage (stage II/III) and 27% for distant-stage (stage IV) disease [6]. This was based on a large-scale analysis of data from the Surveillance, Epidemiology, and End Results (SEER) program in the United States. This dramatic difference in survival underscores the critical importance of early detection, as tumors detected at an earlier, more localized stage are far more treatable and associated with better outcomes [7].

Furthermore, prompt initiation of treatment following diagnosis is also a crucial factor in improving patient survival [7]. Independent of tumor size and other prognostic factors, a study in the Annals of Surgical Oncology found that a delay of more than 60 days between diagnosis and treatment start was associated with a significantly higher risk of mortality [8].

Breast cancer detection relies on a variety of imaging modalities, each with its own strengths and limitations. The most used technique is mammography, which utilizes low-dose X-rays to create images of the breast [9,10]. For decades, mammography has served as the cornerstone of breast cancer screening, demonstrating its effectiveness in detecting early-stage tumors and lowering mortality rates [11]. However, women with dense breast tissue can reduce the sensitivity of mammography, leading to a higher rate of false-negative results [12]. To address the limitations of mammography, healthcare providers often employ additional imaging modalities, such as ultrasound imaging, to create real-time images of the breast, providing valuable information about the composition and structure of suspicious lesions [13]. Magnetic resonance imaging (MRI) [14,15], single photon emission computed tomography (SPECT), and positron emission tomography (PET) are other imaging modalities that can help find cancerous tumors [16].

While these advanced imaging modalities offer improved diagnostic capabilities, they are often more expensive and time-consuming and may not be as widely accessible as traditional mammography [14]. Therefore, there remains a need for alternative, cost-effective, and non-invasive screening methods that can complement or potentially replace the current standard of care [17].

Among the emerging non-invasive techniques, infrared breast thermography has shown promise in early breast cancer detection. Thermography measures temperature variations on the body’s surface and can detect localized temperature increases caused by abnormalities like malignancies, infections, and inflammation [18]. In the case of breast cancer, the temperature of tumor tissue rises by approximately 3 °C compared to the surrounding normal tissue due to the increased metabolic activity of cancer cells [4]. Thermography captures a thermogram of the localized hot spot, which is then interpreted to determine the presence of a breast tumor. Although thermography cannot provide morphological information about the breast, it offers valuable insights into the thermal and circulatory characteristics of the tissue [19]. This makes it a potentially valuable tool for detecting early signs of breast abnormalities before they develop into cancer, as physiological changes occur before pathological changes. Thermography can help reduce unnecessary biopsies for benign breast conditions and identify early cases of breast cancer. It is widely used as a reliable supplementary aid to mammography for early detection of breast cancer [20].

Infrared thermography generates high-resolution heat images of the breast using an ultra-sensitive infrared camera. The camera measures temperature changes on or near the skin surface by detecting infrared radiation emitted from the breast surface, which is then converted into a digital image. It encourages accuracy and fundamental advantages, such as low cost, portability, and painless technique, making it a promising alternative to mammography [17]. Unlike other methods, infrared thermography does not require direct patient contact. The quality and accuracy of the acquired thermal images play a crucial role in the effectiveness of this diagnostic modality. However, the accuracy of this technique in detecting malignancy may be affected by fluctuations in the temperature and humidity of the imaging chamber [21]. Further evaluation is needed to determine the exact role and accuracy of thermography in breast cancer screening.

Researchers have developed various thermal techniques to screen and diagnose breast cancer in its early stages. In recent years, several computer-aided diagnosis (CAD) systems utilizing thermal images have been proposed for breast cancer detection [22]. These systems employ advanced techniques like convolutional neural networks (CNN) and machine learning algorithms to improve classification accuracy [23]. Preprocessing, data augmentation, and careful selection of clinical data have been shown to enhance the performance of these CAD systems. However, further research is needed to evaluate the exact role and accuracy of thermography in breast cancer screening [20,24].

## 2. Related Work

Several studies have focused on breast cancer detection using thermal images and have explored various approaches and methodologies [25,26,27,28]. For instance, a study proposed a computer-aided diagnosis system based on CNN for breast cancer diagnosis using thermal images [25]. The study demonstrated the superiority of CNN-based CAD systems in terms of speed, reliability, and robustness compared to other techniques. Another study developed a CNN machine learning model for breast cancer diagnosis, achieving high accuracy rates by incorporating clinical data choices [26]. The use of deep CNN models and output spectrum analysis has also shown promising results in predicting breast cancer using thermal images [27]. Additionally, self-organizing neural networks and multi-layer perceptron neural networks have been employed for grouping and screening thermal imaging data [28]. These studies highlight the potential of thermography as a cost-effective and less invasive screening option for breast cancer.

For instance, a study proposed a computer-aided diagnosis system based on convolutional neural networks (CNN) for breast cancer diagnosis using thermal images [29]. The study demonstrated the superiority of CNN-based CAD systems in terms of speed, reliability, and robustness compared to other techniques [25]. The results have shown that applying data preprocessing and data augmentation techniques on thermal images leads to lower false positive and false negative classification rates. The CNN models developed in the study achieved a higher accuracy (92%) and F1-score (92%) compared to several state-of-the-art architectures such as ResNet50, SeResNet50, and Inception. Another study developed a CNN machine learning model for breast cancer diagnosis, achieving high accuracy rates by incorporating clinical data choices [26]. This work improves the model using the Visual DMR dataset and clinical data choices. The top model has 85.4% accuracy before clinical data choices and 93.8% after. When categorizing ill individuals as positive, the model had 96.7% specificity and 88.9% sensitivity. The study shows that thermography may be a cheaper, less invasive breast cancer screening option than mammography. The use of deep CNN models and output spectrum analysis has also shown promising results in predicting breast cancer using thermal images [27]. They used a 680-thermogram training dataset to predict breast cancer with 95.8% accuracy using the output spectrum. Additionally, self-organizing neural networks and multi-layer perceptron neural networks have been employed for grouping and screening thermal imaging data [28]. Their suggested breast cancer screening approach has a sensitivity of 88% and 100% and an accuracy of 98.5% for the two case bases.

On the other hand, a recent study [30] explores the use of infrared thermography in medical diagnosis, screening, and disease monitoring. Also, they mention that modern infrared thermal imaging cameras with high-temperature sensitivity and resolution are used in thermography. However, it does not discuss any new technologies or devices in infrared thermography that are specifically related to the detection of breast cancer. Despite these advancements, further research is necessary to evaluate the exact role and accuracy of thermography in breast cancer screening. Our proposed thermal imaging prototype, incorporating compression techniques and advanced classification methods, aims to contribute to this ongoing research and provide a highly effective tool for early breast cancer.

In this paper, we present the development and validation of an innovative thermal imaging prototype that incorporates compression techniques inspired by mammography equipment. The prototype offers a radiation-free and precise diagnosis of breast cancer. By integrating compression and illumination methods, the thermal imaging system enhances the quality of thermal images and improves the accuracy of classification. We designed and constructed a thermography device consisting of essential components such as an equipment body, plates, motors, pressure sensors, light sources, and a thermal camera. A 3D model of the device was created using SolidWorks software, enabling visualization and design optimization.

To evaluate the performance of the proposed system, we conducted extensive experiments using a dataset of cancer and normal thermal images. Preprocessing and segmentation techniques were applied to enhance the quality of the dataset. We employed various classifiers and examined their performance in categorizing the thermal images. The results demonstrated a precision level of 97.6% and high accuracy in detecting breast abnormalities. The proposed prototype serves as an effective tool for initial investigations into breast cancer detection, offering potential advancements in early-stage diagnosis and improving patient survival rates.

In the following sections, the design of the proposed imaging prototype is first detailed. Preprocessing and segmentation techniques applied to both the acquired dataset and a sample of the available breast thermogram datasets are then described. Finally, a classification of the thermal images is performed using various classifiers, and their performance evaluation is discussed.

## 3. Methods and Materials

### 3.1. Design of the Proposed Imaging Prototype

The proposed thermography device is designed to optimize the acquisition and processing of thermal images, aiming to improve the detection and characterization of thermal abnormalities, particularly in the context of breast cancer diagnosis. Integration of compression and illumination techniques synergistically enhances the visibility and quality of thermal images, enabling more accurate interpretation and classification. Figure 1 provides a comprehensive depiction of the proposed thermography device workflow, outlining the specific procedures involved in each stage, namely input, system process, and output. The input stage collects pertinent patient information and prepares the patient for the test. This entails ensuring the patient’s readiness and gathering the necessary details for the next stages.

During the process stage, the patient positions themselves in front of the device and carefully places their breast on the lower plate. Switches facilitate controlled plate movement, allowing for precise alignment of the breast length. To achieve the desired or optimal pressure, the patient manually lowers the upper plate, which incorporates a pressure sensor. To ensure optimal imaging conditions, illuminated LEDs are activated once the plates have been appropriately adjusted based on breast size. This provides the necessary level of brightness for the subsequent capture of thermal images. The system then proceeds to capture thermal images by positioning a fixed thermal camera above the plates. This camera has the capability to rotate around the breast, enabling comprehensive imaging from various angles. We meticulously determine and configure all relevant parameters, such as plate levels, pressure, and LED brightness, prior to image capture.

The output stage acquires and saves a thermal image of the patient’s breast to a computer for subsequent classification analysis. Additionally, we have developed a dedicated software program specifically for thermography equipment. The program receives real-time data from the thermal camera, facilitating advanced analysis and automated cancer detection. The proposed thermography device comprises basic components, equipment body, plates, motors, pressure sensing, illumination sources, and thermal camera. The device components can be further described as follows:(a)Equipment body

The equipment body is constructed using 8 mm acrylic material, known for its strength, stiffness, and optical clarity. The lightweight nature and ease of fabrication of acrylic make it suitable for this application. Furthermore, acrylic exhibits superior weathering properties and impact resistance compared to other transparent plastics. To ensure stability, the device body is supported by a wooden base, which prevents accidental falls. The thermography device stands at a height of 170 cm, with an infrared camera mounted at the top. The design incorporates eight buttons, serving various functions. One button powers the system, while four buttons control the two motors responsible for plate movement, enabling the application of desired pressure. Additionally, three buttons regulate the illumination levels of the LEDs. The uppermost section of the device houses the thermal camera, which captures images upon pressing its dedicated button.

The thermography device receives power from a switching mode power supply (SMPS), which delivers different voltages to meet the power demands of the various system components. These components encompass motors with drivers, an LCD, red LEDs for brightness, and microcontrollers responsible for controlling all aspects of the system. The 3D model of the proposed thermography device was designed using the SOLIDWORKS software program [31] to visualize the device. Figure 2 provides a visual representation of the various components of the proposed thermography device, illustrating its structural layout and functionality.

(b)Plates

The thermography device incorporates a pair of plates to exert mechanical pressure on the object under imaging, enhancing the capture of intricate details. Both plates possess vertical flexibility within a 60 cm range, enabling imaging of patients with varying heights. The colorless acrylic plates, measuring 25 × 25 cm^2^, are equipped with individual motors that facilitate controlled vertical movement. The top plate features 4 mm diameter holes to prevent the reflection of infrared rays during imaging. A visual representation of the top and bottom plates is provided in Figure 3.

(c)Motors, drivers, and controlling system:

The proposed thermography device integrates two stepper motors, each dedicated to the movement of a respective plate. Leadscrews convert the rotary motion of these motors into linear motion (refer to Figure 3). We employ Allegro’s A4988 drivers [20] to control the direction of motion, either upward or downward. These drivers offer adjustable current limiting, over-current and over-temperature protection, and support micro-step resolutions ranging from full-step to 1/16-step. Operating within a voltage range of 8 V to 35 V, the A4988 drivers can provide up to approximately 1 A per phase without the need for additional cooling mechanisms. Each driver enables precise control of one bipolar stepper motor, with an output current capacity of up to 2 A per coil.

The selected stepper motors for the thermography device adhere to standard specifications, delivering a torque of 4.2 kg. Cm with a step angle of 1.8° (equivalent to 200 steps/revolution), these motors consist of four color-coded wires terminated with bare leads: black and green wires connect to one coil, while red and blue wires connect to the other. The control system is responsible for managing the device’s overall operation and coordinating the movement of the two motors that control the upper and lower plates. This controller offers reliable and flexible motor control capabilities combined with a pressure measurement platform. Figure 4 presents a schematic of the controller’s interface with the motors and their drivers.

(d)Pressure sensor:

Breast compression is a vital aspect of mammography examinations, as it immobilizes the breast and reduces tissue thickness for optimal imaging. In our proposed thermography device, we utilize a Force Sensing Resistor (FSR) pressure sensor to measure the applied pressure [32]. The FSR’s resistance changes in response to applied pressure, decreasing as pressure increases. This pressure sensor detects and quantifies the pressure exerted by the mechanical force applied using the lower plate, on which the breast rests, and the upper plate, which applies force to the breast tissue. Several factors, including breast size, skin elasticity, applied force, pain tolerance, and image quality, influence the degree of compression. Conventional breast compression guidelines recommend applying the maximum pressure possible while considering patient pain tolerance and the mammogram unit’s maximum force settings. For an average breast size of 201 ± 55.4 cm^2^, the average force applied in most cases is 110 ± 12.9 N.

(e)Illumination source:

To enhance the imaging of breast tissue, we employ a matrix of red LEDs as an illumination source. These LEDs emit light at a wavelength of 645 nm, making it easier to capture detailed tissue images. We strategically position the LED matrix beneath the lower plate to ensure optimal illumination. Based on the imaged object’s size, we have implemented three levels of illumination. An on/off switch can independently control each illumination level. Figure 5 depicts the arrangement of the LED distribution.

(f)Thermal camera

For capturing thermal images, we have selected an infrared (IR) camera known for its high-resolution image quality [33]. This camera offers a pixel resolution of 160 × 120 for infrared images, a 2.8-inch color display screen, and a thermal sensitivity of 0.07 °C. With a measurement accuracy of ±2 °C or 2%, this camera enables precise temperature measurements. A separate wooden holder mounts the camera, ensuring stability and facilitating image capture from various angles. The vertical distance between the object and the camera ranges from 20 to 55 cm, as indicated in Figure 2, which illustrates the camera components.

There are several imaging considerations related to the device that require adjustment to ensure the acquisition of high-quality images. These adjustments encompass the following:Device distance to examination room light: The proximity of the examination room light to the device can cause unwanted reflections on the surface of the plates, resulting in blurred images. To mitigate this issue, the device must be placed at a sufficient distance, approximately 2 m, from the examination room light or in an indirect location.Camera charging percentage: Adequate charging of the camera is crucial to ensuring that the captured images exhibit suitable resolution. We recommend keeping the camera’s battery level above 30%.Adjust the size of the LED illumination matrix (or level of LED illumination) appropriately to match the imaged breast area, ensuring that it is smaller than the actual breast size. Using a larger illumination area than the breast area can result in overheating of the breast, leading to completely red thermal images, which in turn leads to incorrect data.Camera position: To capture the entire size of the breast, it is important to initially adjust the camera position to be perpendicular to the middle line of the breast. This guarantees the capture of only partial images. The recommended distance between the camera and the breast is within the range of 20–55 cm.

### 3.2. Classification of the Thermal Images

The process of cancer detection may be broadly divided into two main stages: data acquisition and image classification. Data acquisition refers to the procedure of gathering data, whereas picture classification is the categorization of photographs according to their content. In addition, the data acquisition process includes the collection of data, the initial processing of data, the extraction of relevant characteristics, the training of machine learning models, and the generation of inference results. The flow chart for the applied image processing techniques is shown in Figure 6.

#### 3.2.1. Data Acquisition and Dataset Description

We used the proposed thermography device to image a cohort of 48 cases, each with an average age of 12 to 40 years. These cases showed variations in breast area, side, marital status, and child presence. Detailed information regarding a subset of these cases and the corresponding experts’ diagnoses can be found in Table 1. Figure 7 presents a set of thermal images acquired from different angles. The camera’s position varies from the vertical position at 0°, where it aligns with the middle line of the breast, to angles ranging from −15° to +20° with 5° increments. This variation in camera position allows for comprehensive breast imaging from multiple perspectives.

Figure 8 shows a comparison between the acquired thermal images and the corresponding images obtained from different modalities, such as mammograms and ultrasounds, captured from the same viewpoint. The histogram of the collected images, encompassing both normal and abnormal cases, is illustrated in Figure 9.

Various preprocessing and segmentation techniques are applied to prepare the acquired dataset for analysis. These techniques aim to enhance the quality of the thermal images and extract relevant features for subsequent classification [34]. Specific algorithms are employed to correct noise, artifacts, and inconsistencies in the acquired images. Furthermore, segmentation algorithms are utilized to isolate and delineate regions of interest, particularly areas suspected of containing cancerous tissue. The flow chart in Figure 6 illustrates the procedure applied from receiving the thermal image to its classification.

Firstly, the RGB images are read and converted into grayscale images. The next step in preprocessing is contrast enhancement using Gamma correction, as shown in Equation (1), with γ equal to 0.9. Gamma correction involves a non-linear adjustment to individual pixel values:(1)$$O={\left(\frac{I}{255}\right)}^{\gamma }\times 255,$$
where I = the input image; O = the input image; gamma constant controls the shape of the transformation curve. Figure 10 shows the original images and images after applying contrast enhancement.

In this study, we analyzed a homogeneous region of images to assess the nature of the noise present. The histogram of the region provided insights into the type of noise, such as salt and pepper, speckle, Gaussian, etc. In the case of the acquired thermal images, the histogram exhibited a bell-shaped distribution, indicating the presence of Gaussian noise (Figure 11). To effectively address the noise in the acquired images, “Equation (2) shows the Gaussian filter that is the most efficient filter to employ for images”.
(2)$$G_{\sigma }=\frac{1}{2\pi \sigma^{2}}e^{-\frac{\left(x^{2}+y^{2}\right)}{2\sigma^{2}},}$$
where σ is the standard deviation of the noise distribution; (x, y) denotes the position where the filter is applied.

We then evaluated and compared key parameters such as mean square error (MSE) and peak signal-to-noise ratio (PSNR). Our results, presented in Table 2, demonstrate that the Gaussian filter outperformed the other filters. It exhibited the lowest MSE and the highest PSNR, indicating superior performance in noise reduction.

#### 3.2.2. Data Augmentation

Data augmentation is a way of expanding a dataset by modifying the underlying data. In machine learning models, data augmentation is widely used to generate suitable performance in computer vision tasks [35]. The augmentation process was divided into two categories: position and color augmentation. To improve the position, we employed a variety of techniques, including scaling, cropping, shearing modification, padding, flipping, translation, and rotation. For color improvement, we experimented with changing brightness, saturation, and contrast. These augmentation approaches assist in artificially producing fresh training samples by applying various alterations to the source photos. This technique incorporates changes into the training data, making the model able to adjust to diverse orientations, scales, and lighting conditions. The augmented images were merged with the original images, creating a total of 140 images as training samples.

#### 3.2.3. Feature Extraction

Image processing and computer vision tasks are significantly influenced by feature extraction. The extraction of meaningful information from images is crucial for a variety of applications, as they contain enormous quantities of data. Here, feature vectors were obtained by applying the Gabor filter and Gray-Level Co-Occurrence Matrix (GLCM) methods. The feature vectors were dimensionally reduced through the implementation of recursive feature elimination (RFE) and subsequently input into the machine learning models to classify the thermal images as either normal or cancerous. The flow diagram of the feature extraction methods is shown in Figure 12. This combined approach can be a powerful technique for effectively characterizing and leveraging textural information in image-based applications.

##### Gabor Filter

Firstly, we apply the Gabor filter to the images to generate features that represent texture and edges, mimicking the way our eyes recognize texture. The Gabor filter is a collection of bandpass filters that selectively process calculations within a specific frequency range. The utilization of a Gabor filter enables the straightforward identification of edges, textures, and feature extractions by modulating the Gaussian kernel function with a sinusoidal wave [36].

The Gaussian component provides the weights, and the sinusoidal component provides the directionality. The equation that defines the general function for a 2D Gabor filter in the spatial domain is as follows.
(3)$$g\left(x,y:\lambda ,\theta ,\psi ,\sigma ,Y\right)=exp\left(-\frac{x^{\prime 2}+\gamma y^{\prime 2}}{2\times \sigma^{2}}\right)\times exp\left(i\times \frac{2\pi x^{\prime }}{\lambda }+\psi \right)$$
where *x*, *y* represent the position in which we apply the kernel; *x*’ = *x* cos *ϴ* + *y* sin *ϴ*; *y’* = −*x* sin *ϴ* + *y* cos *ϴ*; *λ*: wavelength (# of pixels per cycle); *θ*: orientation of the filter; *Ψ*: phase offset; *σ*: standard deviation; *Υ*: aspect ratio (symmetry of the filter).

To describe the textural properties of masses at different scales and orientations, one can tune Gabor filters with different orientations and scales, forming a Gabor filter bank [24]. Here is the applied feature extraction procedure:Partition each region of interest (ROI) into sub-regions (windows);Separately, apply the Gabor filter bank to each window;Calculate features (mean, standard deviation, and skewness) based on the magnitude of Gabor filter bank responses.

##### Gray-Level Co-Occurrence Matrix (GLCM)

The Gray-Level Co-Occurrence Matrix (GLCM) method is widely used to extract features that consider the interactions between adjacent pixels, providing detailed texture information. GLCM is especially suitable for jobs that involve evaluating intricate texture patterns, such as medical picture analysis. The GLCM method records the spatial arrangement of textures by counting the number of times certain patterns occur in a particular pixel offset (often 1 or 2) and specific orientations (such as horizontal, vertical, or diagonal). After calculating the GLCM, other statistical measures may be obtained from it to describe the texture and structure of the image. Several typical characteristics include contrast, dissimilarity, homogeneity, and autocorrelation. We have applied feature selection to lower the computational cost of modeling and enhance the model’s performance [37].

##### Recursive Feature Elimination (RFE)

RFE is a feature selection technique that recursively removes the least important features until it reaches the desired number of features. By applying RFE, you can optimize the feature set, reducing the computational cost and potentially improving the model’s performance by focusing on the most informative features.

#### 3.2.4. Image Classification

In this study, several machine learning methods were employed to analyze the dataset of thermal images obtained from patients with known breast abnormalities, including tumors. The images were captured using the developed thermography device and subsequently processed through a software system. We divided the dataset into training sets (70%) and testing sets (30%) to train and evaluate the classifiers’ performance, respectively.

Logistic Regression: A linear classification algorithm that models the relationship between the input features and the binary outcomes. It estimates the probabilities of different classes and makes predictions based on a decision boundary [38].

Random Forest: Random Forest is an ensemble learning method that combines multiple decision trees to make predictions. It constructs a multitude of decision trees on random subsets of the data and combines their predictions to achieve higher accuracy and reduce overfitting [39].

Gradient Boosting: It is another ensemble learning technique that sequentially builds weak prediction models and focuses on samples that were previously misclassified. It combines the predictions of multiple weak models to create a strong predictive model [40].

AdaBoost: AdaBoost, short for Adaptive Boosting, is an ensemble learning algorithm that iteratively trains weak models on different weighted versions of the dataset. It assigns higher weights to misclassified samples in each iteration, allowing subsequent models to focus more on these samples and improve overall accuracy [41].

Neural Networks: The neural network architecture consists of interconnected layers of artificial neurons, which process and transform the input features to generate the target predictions [42]. The input layer uses a feature vector consisting of 30 reduced features. We used the Adam activation function on a total of 50 hidden layers. The ultimate output layer utilizes a linear activation function to generate the predicted target variable on the appropriate scale. The neural network model is trained using the Adam optimization algorithm [43], with a learning rate of α = 0.0001. We iterate the training process for 200 epochs, monitoring the training process by tracking the evolution of the loss function over the training iterations.

We applied these machine learning classifiers to the processed thermal images to classify them as either normal or abnormal, indicating the presence of breast abnormalities such as tumors. We divided the dataset into training and testing sets to train the classifiers on a subset of the data and assess their performance on unseen data. This approach helps assess the ability of the classifiers to generalize to new instances and make accurate predictions on unseen thermal images.

## 4. Results and Discussion

This research study conducted an experimental evaluation to assess the performance of the proposed thermography device. The experimental results demonstrate the efficacy of the thermography device in capturing high-quality thermal images with enhanced details. Notably, the inclusion of mechanical pressure plates in the device facilitated better breast tissue visualization and characterization.

Figure 13 provides a visual representation of the thermal images acquired with and without the compression technique. The application of compression resulted in a flattened breast surface, enabling comprehensive illumination of all breast regions along with the complete contour of the outer dimension.

The utilization of compression techniques inspired by mammogram machines, coupled with the elimination of ionizing radiation, presents a safer and more accessible alternative for breast cancer detection. Careful selection of the lighting source ensures proper illumination of the object and facilitates the acquisition of high-quality thermal images using the thermal camera. The developed device successfully captured the thermal images, which a software system then processed.

The preprocessing techniques implemented in this study played a crucial role in removing noise and artifacts from thermal images, thereby enhancing the accuracy of the subsequent classification.

The proposed thermography device’s accuracy and reliability in classifying thermal images, as well as its efficiency in producing valuable thermal images for cancer detection, are noteworthy. The prototype’s integration of compression techniques and illumination methods ensures that the acquired thermal images have improved quality and clarity. This enhanced image quality plays a crucial role in detecting subtle thermal abnormalities associated with breast cancer. By capturing and representing thermal patterns with higher fidelity, the proposed thermography device provides valuable visual information for clinicians and researchers involved in cancer detection.

The device’s efficiency in producing high-quality thermal images improves the overall effectiveness of the classification process, resulting in more accurate and confident diagnoses. Furthermore, the radiation-free nature of the device ensures patient safety while delivering reliable thermal images, making it a valuable tool in the early detection and monitoring of breast cancer.

In addition, this study performs a comprehensive classification of the acquired thermal images from the proposed thermography device using a range of classifiers and evaluates their performance. We trained the classification models on a labeled dataset (cancer and normal images), which included expert-annotated ground truth images, to ensure the reliability and accuracy of the classification process. In our study, we compared the performance of five machine learning models, namely logistic regression, Random Forest, Gradient Boosting, AdaBoost, and a neural network.

To evaluate the performance of the different implemented models, the test dataset is utilized to assess the model’s accuracy and effectiveness in classifying images. The models’ performance is assessed by calculating the following metrics:

Classification Accuracy (CA) measures the overall correctness of the model’s predictions.

F1-score is the harmonic mean of precision and recall, providing a balanced measure of the model’s performance.

Precision is the proportion of true positive predictions out of all positive predictions made by the model.

Recall is the model’s proportion of true positive predictions out of all actual positive instances.

The results show that the logistic regression model achieved the highest performance across all metrics, with a classification accuracy of 0.976, an F1-score of 0.977, a precision of 1.000, and a recall of 0.995. This indicates that the logistic regression model was able to make highly accurate and consistent predictions with a low rate of false positives and false negatives. This indicates a high level of accuracy in correctly classifying thermal abnormalities associated with breast cancer.

The neural network model also performed well, with a classification accuracy of 0.947, an F1-score of 0.943, a precision of 0.953, and a recall of 0.932. This suggests that the neural network was able to effectively learn the underlying patterns in the data and make reliable predictions.

In comparison, the Random Forest, Gradient Boosting, and AdaBoost models showed lower but still reasonably excellent performance, with classification accuracies ranging from 0.777 to 0.883. The Random Forest model had the second-highest performance, with an F1-score of 0.864, precision of 0.946, and recall of 0.795. The detailed performance metrics of all proposed models are provided in Table 3.

The results of the performance evaluation provide valuable insights into the effectiveness of the proposed thermography device in detecting and classifying thermal patterns indicative of breast cancer.

Figure 14 displays the confusion matrix for each classifier as a table. The rows of the matrix represent the true classes, and the columns represent the predicted classes. Each cell in the matrix represents the percentage of instances that fall into a particular combination of true and predicted classes. Additionally, we plot a comparative ROC curve for various classifiers in Figure 15, with the *x*-axis representing the false positive rate (FPR) and the *y*-axis representing the true positive rate (TPR).

Validating the classification models against expert-annotated ground truth images is another way to make sure that the proposed thermography device is reliable and accurate. The study shows that the proposed device can correctly identify thermal abnormalities linked to breast cancer by comparing the predicted classifications with the ground truth labels. This helps with early detection and diagnosis.

It is important to acknowledge that further research and evaluation are necessary to validate the proposed thermography device on larger and more diverse datasets. Additionally, future studies could investigate the integration of other machine learning techniques and feature extraction methods to improve the device’s classification performance.

In conclusion, the results of this study demonstrate that the proposed thermography device, combined with the implemented classification models, shows promising potential for accurately identifying and classifying thermal abnormalities associated with breast cancer. The models’ high level of accuracy and precision shows how important thermography is as an extra tool for finding and diagnosing breast cancer early. This could cut down on unnecessary biopsies and improve patient outcomes.

## 5. Conclusions

This study’s findings highlight the successful implementation of the proposed thermography device for capturing detailed and informative thermal images. The use of mechanical pressure plates and compression techniques contributes to improved breast tissue visualization and characterization. Notably, the inclusion of mechanical pressure plates in the device facilitated better breast tissue visualization and characterization. This device shows promise as a radiation-free and accessible tool for breast cancer detection, offering a valuable alternative to traditional modalities. The captured thermal images provide a reliable basis for further analysis and interpretation, supporting the early detection and monitoring of breast abnormalities. The study concludes that the proposed thermography device, when paired with the implemented classification models, exhibits promising potential in accurately identifying and classifying thermal abnormalities linked to breast cancer. The models’ high level of accuracy and precision shows how important thermography is as an extra tool for finding and diagnosing breast cancer early. This could cut down on unnecessary biopsies and improve patient outcomes.

## Figures and Tables

**Figure 1 bioengineering-11-00764-f001:**
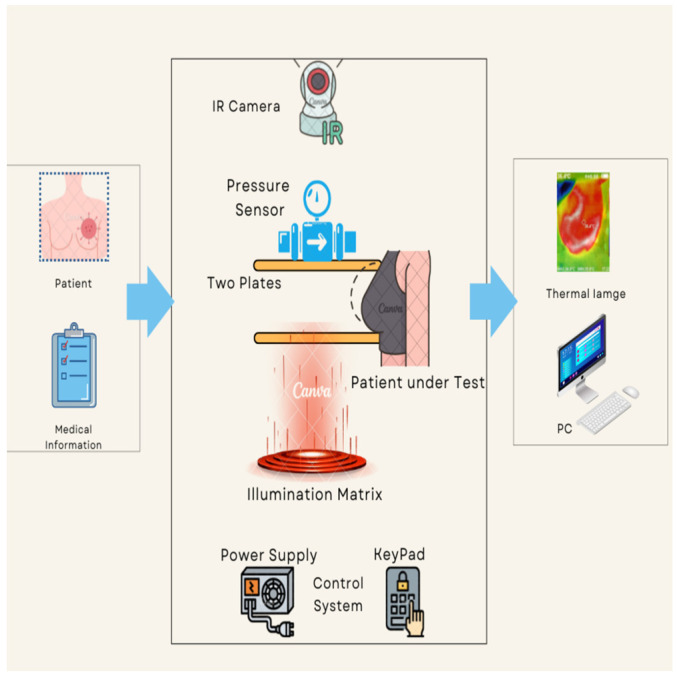
The proposed thermography device diagram.

**Figure 2 bioengineering-11-00764-f002:**
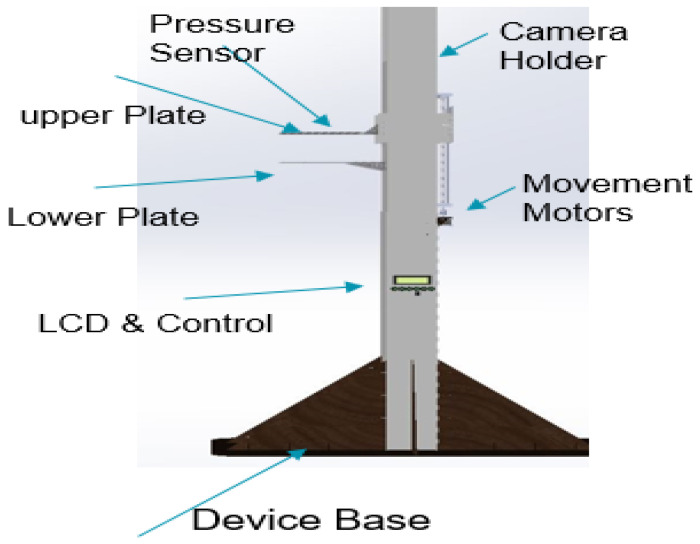
Thermography device parts.

**Figure 3 bioengineering-11-00764-f003:**
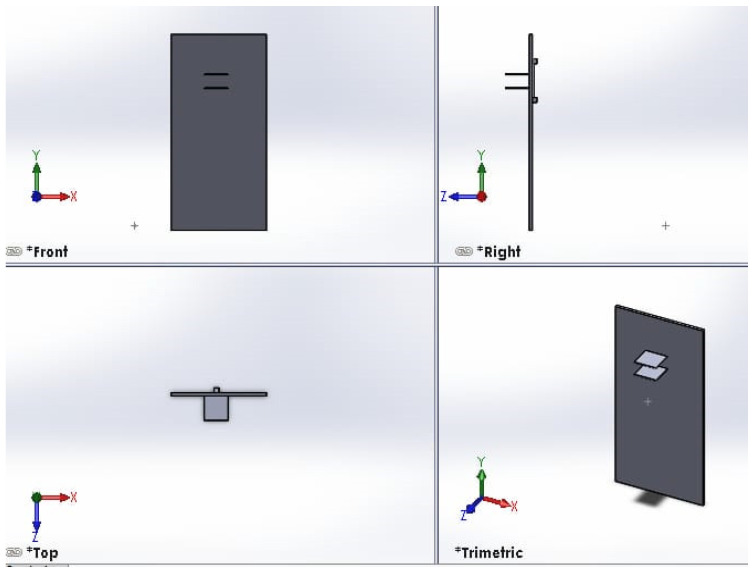
Front view of top and bottom plates. The right view shows the side view of the motor for moving one plate. A visual representation of the top and bottom plates, created using the SOLIDWORKS software program. The figure is divided into four views: front, right, top and trimetric. The front view shows the overall layout and relative positioning of the top and bottom plates. The right view shows the side view of the motor used for moving one of the plates. The top view provides important details on the dimensions and geometry of the plate components and the motor used for moving one of the plates. The trimeric view offers a three-dimensional perspective that helps visualize how the top and bottom plates fit together and interact.

**Figure 4 bioengineering-11-00764-f004:**
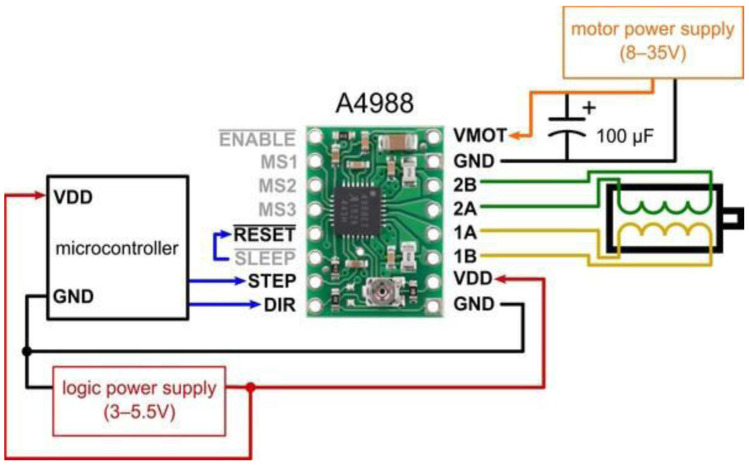
Connection diagram for controller interfacing with the motor and its driver.

**Figure 5 bioengineering-11-00764-f005:**
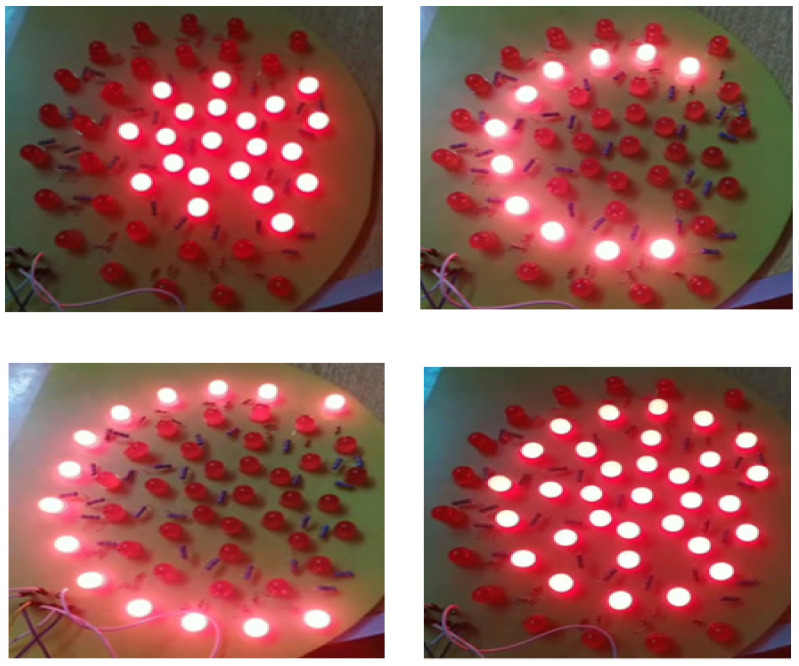
The arrangement of illumination lights based on the selected level and breast size.

**Figure 6 bioengineering-11-00764-f006:**
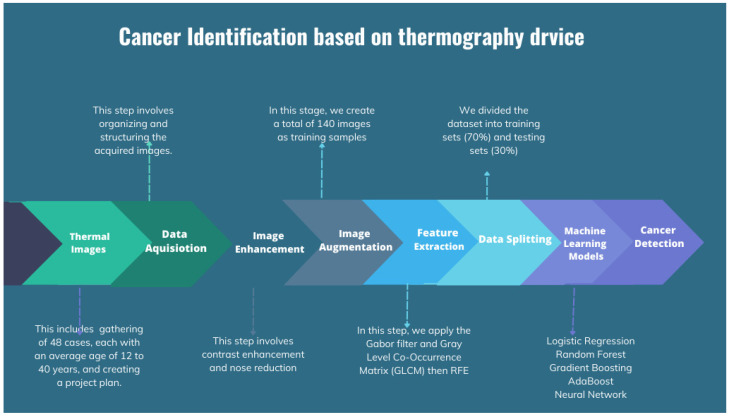
Flow chart for the applied image processing techniques.

**Figure 7 bioengineering-11-00764-f007:**
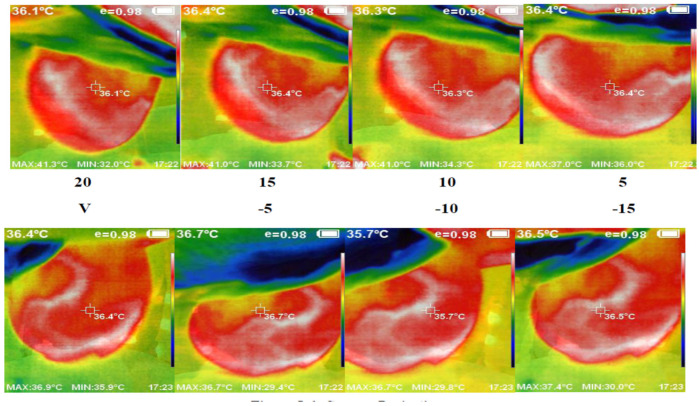
Acquired abnormal right breast image in the different angles via camera (8 images from 5 to 20 °C).

**Figure 8 bioengineering-11-00764-f008:**
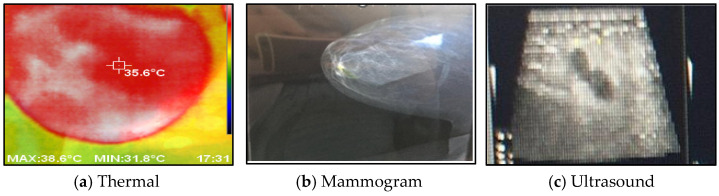
The acquired thermal images versus other imaging modalities.

**Figure 9 bioengineering-11-00764-f009:**
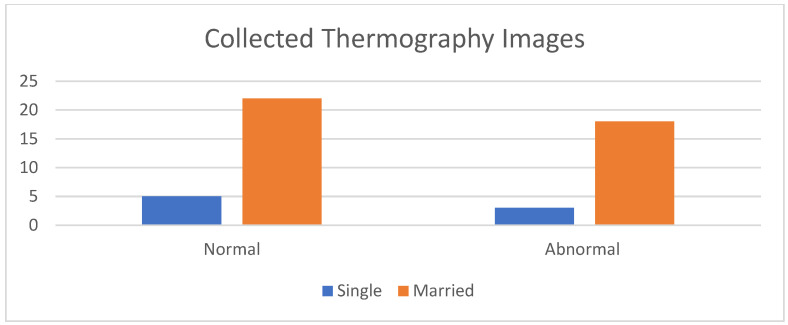
Histogram of the collected data.

**Figure 10 bioengineering-11-00764-f010:**
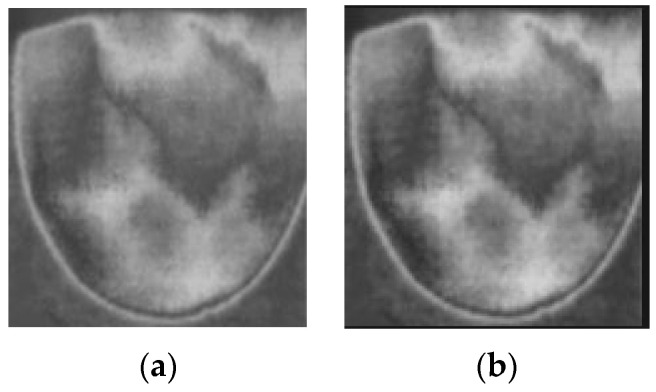
Applying contrast enhancement to a sample image. (**a**) Before contrast enhancement; (**b**) after contrast enhancement.

**Figure 11 bioengineering-11-00764-f011:**
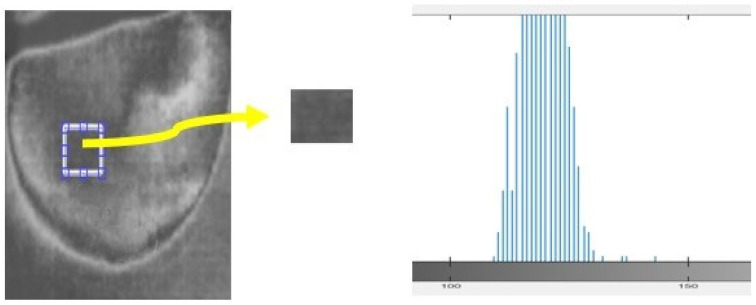
Histogram calculated for samples of the acquired thermal image.

**Figure 12 bioengineering-11-00764-f012:**
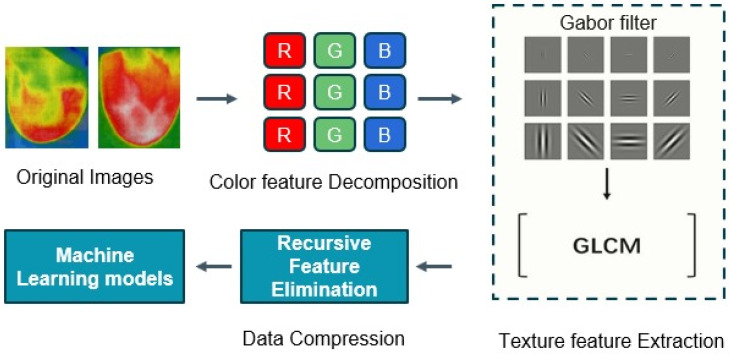
The flow diagram of the feature extraction methods.

**Figure 13 bioengineering-11-00764-f013:**
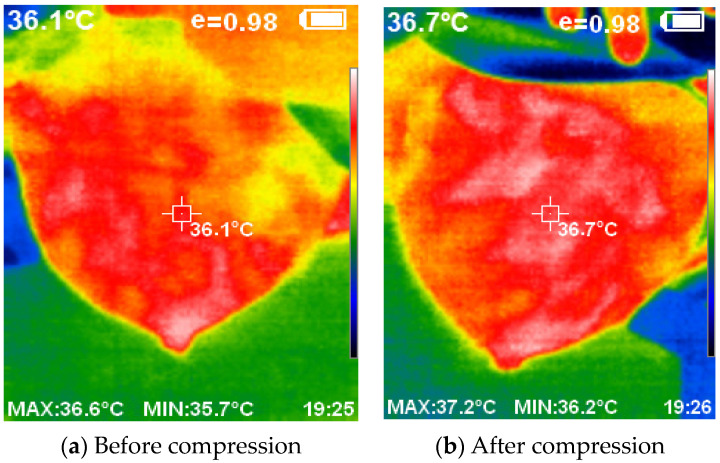
The acquired thermal images using and without compression.

**Figure 14 bioengineering-11-00764-f014:**
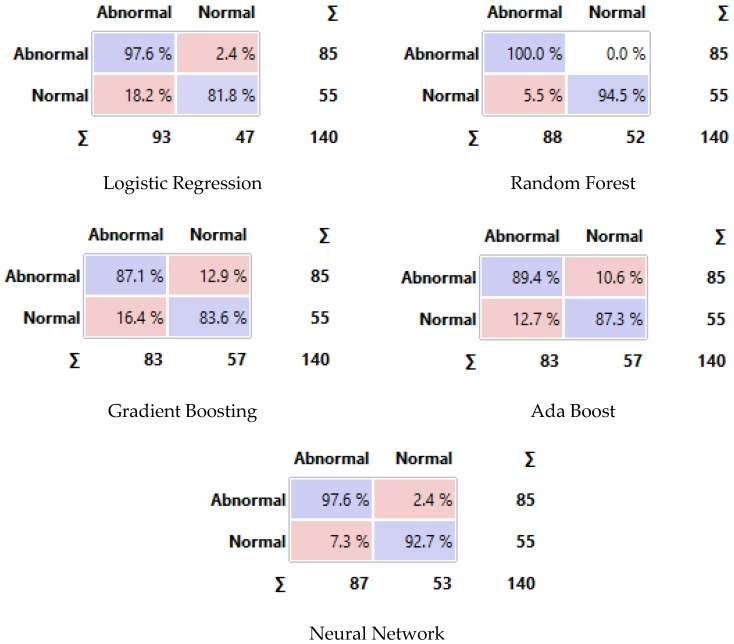
Confusion matrix of different classifiers. The Purple color represents the correct predictions and the pink color for false predictions.

**Figure 15 bioengineering-11-00764-f015:**
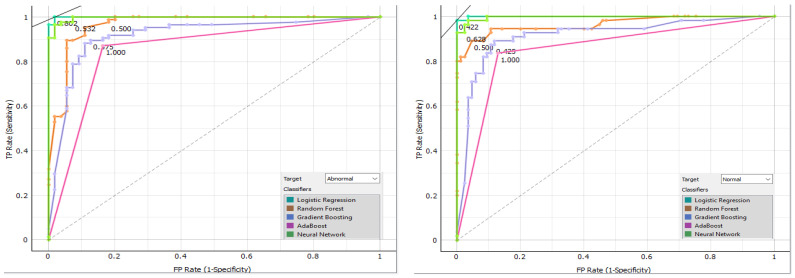
Comparative ROC curve for various classifiers.

**Table 1 bioengineering-11-00764-t001:** Example of patient information and the corresponding experts’ diagnosis.

Case No.	Age	Imaged Breast	Marital Status	Presence of Children	Imaging Modalities	Experts’ Diagnosis
1	33	Left	Married	1	No	Abnormal
2	39	Left	Married	2	Mammo	Normal
3	39	Right	Married	1	Mammo	Abnormal
4	41	Right	Single	0	No	Normal
5	46	Left	Married	1	No	Normal
6	40	Right	Married	0	Mammo, US	Abnormal
7	38	Left	Single	0	No	Normal

**Table 2 bioengineering-11-00764-t002:** Performance analysis of the applied filters.

Performance Measure	Gaussian	Weiner	Median
Mean square error (MSE)	0.218	6.35	62.72
Peak signal-to-noise ratio (PSNR)	54.735	40.09	30.156

**Table 3 bioengineering-11-00764-t003:** Evaluation of classifiers based on performance metrics.

Model	CA	F1-Score	Precision	Recall
Logistic Regression	0.976	0.977	1.000	0.995
Random Forest	0.883	0.864	0.946	0.795
Gradient Boosting	0.830	0.826	0.792	0.864
AdaBoost	0.777	0.769	0.745	0.795
Neural Network	0.947	0.943	0.953	0.932

## Data Availability

The database that supports the results of this research is collected via Benha University Hospital. We were concerned only with data analysis and methodology and not with any clinical testing. This study has been approved by the medical ethics committee (Ethical Statement).

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
