# Peer review of "An Innovative Thermal Imaging Prototype for Precise Breast Cancer Detection: Integrating Compression Techniques and Classification Methods"

_bioengineering, 2024, doi:10.3390/bioengineering11080764_

Round 1
Reviewer 1 Report
Comments and Suggestions for Authors
The problems with the paper are as follows:
1.Image enhancement was performed on Figure 10, and after enhancement, there is no visual difference between Figure 10b) and Figure 10a). It is recommended that the author provide a reasonable explanation.
2.Figure 6 is too macro level, it is recommended to further refine and provide specific implementation methods for each step.
Comments on the Quality of English Language
Minor editing of English language required.
Reviewer 2 Report
Comments and Suggestions for Authors
The introduction contains references to 17 publications, which is clearly not enough for the problem under consideration, which is the subject of many works.
429: It is necessary to explain why (3) has a periodic solution along the coordinate $x$, and not along $y$.
458: The description of the Recursive Feature Elimination method must be detailed.
Each method in section 3.2.3 must be supported by detailed material (graphs, diagrams, formulas, etc.).
The results of machine learning are not well described and require evidence-based explanation.
All object images need color or gray gauge bars. Quantitative characteristics must be present.
It is not clear why Fig. 12b is better than 12a? The upper right part of these drawings is radically different. This difference is so great that it is as if we are seeing different data.
Using the Neural Network requires a description of the architecture, plotting the dependence of the error on the training epoch, and a discussion of the problem of overtraining.
The description of Table 3 is insufficient. Simply listing that high accuracy, precision, and recall scores have been achieved cannot be considered a scientific result.
279: Should be "cm$^2$".
373: The formula is part of a sentence and all punctuation rules apply to such sentences (commas, red line, etc.).
429: Likewise.
385: A sentence cannot begin with a formula and a sentence cannot consist only of a formula. All quantities in all formulas must be defined immediately after the formula.
389: Should be “Before” and there is an extra space before the comma.
427: Remove "(equation 3)" from line 427.
429: The right side of equation (3) does not contain $\theta$, $Y$.
What is $pi$ ? Is this $\pi$?
430-436: The standard definition of the quantity after the formula looks, for example, like this
"$\sigma$ is the standard deviation,". Must be submitted as part of a single proposal. All mathematical symbols must respect roman or italic style. The meaning of the quantity depends on whether roman or italic is used.
442: Define the abbreviation "ROI".
448: What is GLCM?
Comments on the Quality of English LanguageModerate editing of English language required.
Round 2
Reviewer 1 Report
Comments and Suggestions for Authors
Accept in present form.
Comments on the Quality of English LanguageMinor editing of English language required.
Author Response
Dear Esteemed Reviewer,
Thank you so much for your help in enhancing our manuscript.
I appreciate your acceptance of the manuscript.
Regarding minor English editing suggestions;
The paper is revised using Grammarly Premium and QuillBot Paraphrasing Tool.
Reviewer 2 Report
Comments and Suggestions for Authors
The presented version of the manuscript makes it extremely difficult to analyze the quality of the revision in accordance with each comment. It is unclear what comments were processed and how they were taken into account in the text.
Even simple comments related to the design of sentences with formulas have not been finalized.
The standard approach is to have a table in the form:
Reviewer's Note 1 |
Answer: where and what has been changed. Or an argument that challenges the remark. |
Reviewer's Note 2 |
Answer: where and what has been changed. |
… |
|

Comments on the Quality of English LanguageMinor editing of English language required
Author Response
Dear Esteemed Reviewer,
Thank you so much for your help in enhancing our manuscript.
We apologize for any confusion during.
Kindly find the attached file in the same format as you request.
